# The Compromised Intestinal Barrier Induced by Mycotoxins

**DOI:** 10.3390/toxins12100619

**Published:** 2020-09-28

**Authors:** Yanan Gao, Lu Meng, Huimin Liu, Jiaqi Wang, Nan Zheng

**Affiliations:** 1Key Laboratory of Quality & Safety Control for Milk and Dairy Products of Ministry of Agriculture and Rural Affairs, Institute of Animal Sciences, Chinese Academy of Agricultural Sciences, Beijing 100193, China; gyn758521@126.com (Y.G.); menglu@caas.cn (L.M.); liuhuimin02@caas.cn (H.L.); 2Laboratory of Quality and Safety Risk Assessment for Dairy Products of Ministry of Agriculture and Rural Affairs, Institute of Animal Sciences, Chinese Academy of Agricultural Sciences, Beijing 100193, China; 3Milk and Milk Products Inspection Center of Ministry of Agriculture and Rural Affairs, Institute of Animal Sciences, Chinese Academy of Agricultural Sciences, Beijing 100193, China; 4State Key Laboratory of Animal Nutrition, Institute of Animal Sciences, Chinese Academy of Agricultural Sciences, Beijing 100193, China

**Keywords:** mycotoxins, intestinal barrier, intestinal inflammation, interactive effects

## Abstract

Mycotoxins are fungal metabolites that occur in human foods and animal feeds, potentially threatening human and animal health. The intestine is considered as the first barrier against these external contaminants, and it consists of interconnected physical, chemical, immunological, and microbial barriers. In this context, based on in vitro, ex vivo, and in vivo models, we summarize the literature for compromised intestinal barrier issues caused by various mycotoxins, and we reviewed events related to disrupted intestinal integrity (physical barrier), thinned mucus layer (chemical barrier), imbalanced inflammatory factors (immunological barrier), and dysfunctional bacterial homeostasis (microbial barrier). We also provide important information on deoxynivalenol, a leading mycotoxin implicated in intestinal dysfunction, and other adverse intestinal effects induced by other mycotoxins, including aflatoxins and ochratoxin A. In addition, intestinal perturbations caused by mycotoxins may also contribute to the development of mycotoxicosis, including human chronic intestinal inflammatory diseases. Therefore, we provide a clear understanding of compromised intestinal barrier induced by mycotoxins, with a view to potentially develop innovative strategies to prevent and treat mycotoxicosis. In addition, because of increased combinatorial interactions between mycotoxins, we explore the interactive effects of multiple mycotoxins in this review.

## 1. Introduction

Mycotoxins are the non-enzymatic poisonous metabolites produced by fungi such as *Aspergillus, Penicillium,* and *Fusarium* genera [1,2]. In recent years, approximately 500 mycotoxins derived from these fungal species and others have been identified [3]. Of these, considerable attention has been given to several common mycotoxins, which affect both human and animal health, as well as economic growth. These include aflatoxins (AFs), ochratoxin A (OTA), deoxynivalenol (DON), fumonisins (FBs), zearalenone (ZEN), patulin (PAT), nivalenol (NIV), and citrinin (CTN) [4,5]. In terms of carcinogenicity, mycotoxins have been classified into five groups by the International Agency for Research on Cancer (IARC). AFB1 and AFM1 are categorized as Group 1, which reflects their human carcinogen status. OTA and FB1 are classified as Group 2B carcinogens, whereas, DON, ZEN, PAT, and NIV are assigned to Group 3. The toxic effects, classification, and health guidance values are summarized in Table 1. 

Mycotoxins are believed to be present throughout the food chain and likely occur in raw crops and crop by-products [11,12]. If livestock consume mycotoxin-contaminated ingredients, these may become distributed in their meat [13], eggs [14], and milk [15]. Hence, a predominant source of human mycotoxin exposure is derived from the ingestion of contaminated foodstuffs. Following consumption, the upper part of the intestine absorbs these mycotoxins [16,17]. Therefore, the intestinal epithelial barrier represents the first defensive barrier towards mycotoxins, suggesting this organ is more than likely to exposed to higher mycotoxin concentrations than other tissues [18,19]. The intestine is the foremost target organ of mycotoxin toxicity, with its primary role maintaining intestinal homeostasis. It is therefore vital to understand the compromised intestinal barrier mechanism induced by mycotoxins.

## 2. Components of the Intestinal Barrier

The intestinal barrier is composed of interconnected physical, chemical, immunological, and microbial barriers (Figure 1). As indicated, barrier function depends on the dynamic interaction of luminal microbiota, epithelial cells, and immune cells in the lamina propria (LP). A single-cell epithelial layer covers the gut wall and is pivotal to maintaining this physical barrier [20]. The intestinal epithelium is composed of four major intestinal cell types: (i) absorptive enterocytes, which account for the main cell complement, at >80% of the epithelium; (ii) protective mucin-producing goblet cells; (iii) Paneth cells, which produce antimicrobial peptides (AMPs); (iv) hormone-secreting enteroendocrine cells [21]. Epithelial cells are connected to each other from the basolateral to apical direction through interconnected protein contacts (apical junctional complex), which consist of desmosomes, adherens junctions (AJs), and tight junctions (TJs) [22]. Molecules permeate this intestinal epithelium via transcellular and paracellular routes, the latter being regulated by TJs [23,24]. Therefore, TJs selectively regulate nutrients and stimuli flux and are considered the foremost determinants of intestinal paracellular permeability [25]. TJs are multi-protein complexes, and they consist of different transmembrane proteins, e.g., junctional adhesion molecules (JAMs), claudins, and occludin and zonula occludens (ZO) proteins [26,27]. 

As depicted (Figure 1), the term chemical barrier refers to the mucus layer of antimicrobial-related proteins (e.g., mucins and AMPs) secreted by intestinal epithelial cells [21,28]. This barrier prevents luminal bacteria from coming into direct contact with the intestinal epithelium [29]. Mucins are the main constituents of the mucus’ thick matrix and are produced and secreted by goblet cells [30,31]. Mucin 2 (MUC2) is the most abundant protein in the intestine and plays an important role in mucus layer integrity and function [30]. In mice lacking MUC2, luminal bacteria come into direct contact with epithelial cells, thereby inducing inflammation-related diseases such as spontaneous colitis and ulcerative colitis (UC) [32]. AMPs are primarily secreted by Paneth cells, which are only located in the small intestine, while low AMP levels are secreted by enterocytes [28]. 

The immunological barrier comprises immune cells in LP (e.g., dendritic cells, resident macrophages, B cells, and T cells) and the secreted immune mediators (e.g., secretory immunoglobulin A (sIgA) and cytokines) [23,33,34]. SIgA is generated by B cells in the LP, transposed across the epithelium, and secreted into the mucus layer [35]. Cytokines are the important regulators in inflammatory and immune responses, and they are secreted by immune cells and epithelial intestinal cells [18].

More than 10^14^ commensal bacteria, known as the microbiota, comprising hundreds of different species, colonize the mammalian intestinal tract to form the intestinal microbial barrier. This microbial ecosystem plays a key role in maintaining intestinal health, as microbial dysbiosis leads to intestinal inflammation [36,37,38]. To comprehensively understand mycotoxin-mediated damage to the intestine, we systematically analyzed dysfunction mechanisms related to these physical, chemical, immunological, and microbial barriers.

## 3. Experimental Models Used to Assess the Intestinal Barrier

Several in vitro, ex vivo, and in vivo approaches have been used to recapitulate animal intestinal environments to assess and characterize the intestinal barrier (Table 2). 

### 3.1. In Vitro Intestine Model

In vitro models, representing the intestinal barrier, commonly refer to intestinal epithelial cells cultured in transwell chambers with a semipermeable membrane filter. They are typically referred to as two-dimensional (2D) cultures. To date, several intestinal epithelial cell lines from different animal species and humans have been generated. Among these, the porcine intestinal epithelial cell lines, IPEC-1 and IPEC-J2, and the well-established human Caco-2 cell line are the most frequently used intestinal models for studying barrier functions [39,44]. After seeding onto transwell chambers for 1–3 weeks, these cell lines spontaneously differentiate to form polarized monolayers, representing in vitro intestinal physical and chemical barriers [52,53,54]. In addition to these models, IPI-2I and PSI-1 cells from the adult boar ileum and adult pig small intestine, respectively, have also been used as intestinal barrier models [40,55]. However, in considering the complexity interaction between different cell types, monocultures may only partially represent the intestinal barrier. Therefore, multi-culture systems are often preferred [56]. The co-culturing of polarized IPEC-J2 cells with porcine peripheral blood mononuclear cells (PBMCs), and the co-culture of Caco-2 cells and intestinal HT29-MTX cells, has been successfully applied to evaluate intestinal barrier modulation by mycotoxins [42,43,57]. While these aforementioned cell lines are commercially available, freshly isolated intestinal primary cells from various animal species are considered as more accurate models in mimicking in vivo physiology. However, they fail to achieve long-term culture. Therefore, different cell lines are chosen based on specific research purposes.

In addition to the traditional approach of 2D intestinal epithelial cell monolayers, 3D-intestinal models (enteroids), also known as “organoids” or “mini-guts”, which are derived from individual intestinal stem cells, have been developed [41,47]. Currently, based on the enteroids technique of human and mouse models, 3D culture systems have been established for various species including pig, chicken, cow, sheep, and horse [58]. Furthermore, enteroids can be cultured in a multitude of tissues including liver, stomach, colon, etc. [59]. These systems are advantageous for the following reasons: (i) enteroids harbor most, if not all, intestinal cell types [48]; (ii) it is almost impossible to accumulate mutations; (iii) enteroids can be passaged to an almost unlimited scale [45,48,49]. Although enteroids are regarded as a new exploratory intestinal model, they have several limitations: (i) when compared with the well-established and relatively cheap 2D approaches, enteroid systems are considerably costly [41,46]; (ii) the effects of molecules on the enteroid luminal side are poorly investigated [49]. To circumvent this latter hurdle, a combinatorial 2D culture and 3D enteroid approach was developed, where the enteroids were dissociated into single-cell suspensions and then cultured in the transwell chambers [60]. When compared with conventional 2D cultures, 3D enteroids partially recapitulate the anatomy of native epithelium. If researchers require better models to simulate the intestine, and the corresponding technology is available, then 3D enteroid approaches are the best choice.

### 3.2. Ex Vivo Intestine Model

Ex vivo studies involve the isolation of living functional tissue or organs from an organism and culturing in an Ussing chamber [61]. This model has been applied to various species, including humans [50], rodents [62], swine [51], poultry [51], and horses [63]. A main advantage of this intestinal explant culture systems is that the model maintains the histological architecture and complex intestinal barrier integrity under in vivo conditions [64]. Similarly, researchers can isolate specific segments of the gut, from the duodenum to colon, that could answer their research questions [61]. Small bowel explant cultures have been successfully used for celiac disease studies [65], and colon explants investigations have been powerful in studying the effects of intestinal microorganisms [66,67]. However, major drawbacks exist for intestinal explants, thereby restricting their application. These include short-term preservation during culture and careful and laborious preparation [39,61]. While these limitations are technical in nature, ex vivo intestinal models are widely used to evaluate gastrointestinal toxicity induced by mycotoxins [68,69,70,71,72].

### 3.3. In Vivo Intestine Model

In vivo studies, based on information from whole-animal systems, can effectively emulate the toxicity mechanisms in humans. The role of live animals for education and research purposes has been recognized by the World Organisation for Animal Health (OIE) [73]. In general, several animal species, such as mice, rats, chickens, turkey, fish, pigs, sheep, and cows, have been used to evaluate the toxicology of mycotoxins. When choosing a particular animal model for toxicological research, it must be borne in mind that animal species often show different susceptibilities to mycotoxins [1,74]. In much of the contemporary research on mycotoxicology, DON has been the most frequently analyzed mycotoxin. Researchers have observed that traditional in vitro systems, based on cell models, cannot comprehensively characterize pharmacokinetics, bioavailability, and in vivo metabolism, thus necessitating the requirements for in vivo animal models [75]. However, there are the drawbacks to such experimentation (i.e., time-consuming, expensive, and individual differences cannot be ignored). Additionally, from an ethical and animal welfare perspective, the use of live animals should follow the 3R (replacement, reduction, and refinement) principle. Therefore, toxicological assessments cannot be completely dependent on animal tests. Critically, experimental models should be selected based on the scientific area, e.g., in vivo models focus on the overall changes in an organism, while in vitro model investigations are ideal if particular pathways are affected by mycotoxins.

## 4. Intestinal Dysfunction Induced by Mycotoxins

Figure 2 shows information on an intestinal barrier compromised by mycotoxins, including disrupted intestinal integrity (physical), thinned mucus layer (chemical), imbalanced inflammatory factors (immunological), and dysfunctional bacterial homeostasis (microbial). In the following sections we discuss the effects of common mycotoxins such as AFs, OTA, DON, ZEN, FB1, PAT, and CTN.

### 4.1. Effects of Mycotoxins on the Physical Barrier

Numerous studies have shown that mycotoxins disrupt the intestinal physical barrier (Table 3 and Table 4). In general, intestinal epithelial cells are self-renewed every 3–5 days [76], except for Paneth cells, which undergo renewal every 18–23 days. Thus, rapidly proliferating and regenerating intestinal epithelial cells maintain the intestinal physical barrier. Besides these cells, TJ proteins also play roles in the intestinal physical barrier. A previous study has reviewed the effects on intestinal permeability induced by mycotoxins [34]; therefore, in the present study, we will pay more attention to the latest research.

#### 4.1.1. Effects of Mycotoxins on the In Vitro Physical Barrier 

Several mycotoxins modulate the intestinal physical barrier by affecting in vitro intestinal cell proliferation. AFB1 and AFM1 significantly inhibited cell growth, causing cell cycle arrest in Caco-2 cells [77,78,79,80]. In addition, AFM1 and OTA inhibited the cell viability of the co-culture of Caco-2/HT29-MTX [43]. Equally, OTA promoted apoptosis in IPEC-J2 and Caco-2 cells by inducing mitochondrial reactive oxygen species (ROS) and arresting the cell cycle [81,82,83]. Apart from mycotoxin effects in human intestinal cells, DON inhibited intestinal epithelial cell viability and induced apoptosis in rat and pig cell models [84,85,86,87,88]. ZEN-induced cell death was confirmed in IPEC-J2, IPEC-1, and human colon carcinoma cells (HCT116 cells) [89,90,91,92]. FB1, PAT, and CTN induced cell death and apoptosis in the human colon proliferating intestinal cell line (HT-29), Caco-2 cell, and HCT116 cell [93,94,95,96,98,126,127]. These studies showed that mycotoxins caused intestinal epithelial cell death, resulting in damage to the intestinal physical barrier. 

Apart from cell proliferation, TJ-mediated intestinal permeability is also modulated by mycotoxins. AFM1 and OTA exposure to both apical and basolateral surfaces increased intestinal permeability, reduced TJ proteins protein expression levels, and affected the TJ protein distribution pattern [54]. These TJ protein localization effects were confirmed by transmission electron micrographs [57]. OTA treatment resulted in intestinal barrier dysfunction, which was confirmed by increased cell permeability and microvilli disruption and TJ proteins in various cell culture systems [82,105]. These observations could be explained by ROS/Ca^2+^-mediated myosin light chain kinase (MLCK) activation [128]. Exposure to DON decreased transepithelial electrical resistance (TEER) values and the abundance of TJs in a concentration- and time-dependent manner [106,108,109]. It has been reported that ZEN has no effect on TEER values of IPEC-1 cell, while its metabolites (α-ZOL and β-ZOL) led to time-dependent decreased TEER values [118]. FB1 and its metabolite, hydrolyzed FB1 (HFB1), damaged the intestinal integrity in different intestinal cell systems [42]. Previous studies have shown that PAT and T-2 toxin increased differentiated Caco-2 cells permeability, via the destruction of TJs, and was accompanied by MLC2 phosphorylation [94,95,122,124]. These results indicated that TJ expression levels are associated with TEER values and potentially represent a change in intestinal permeability.

#### 4.1.2. Effects of Mycotoxins on the Ex Vivo Physical Barrier 

An increasing number of ex vivo studies have determined the effects of mycotoxins on the intestinal physical barrier. Recently, DON was reported to inhibit growth and reduce ZO-1 and claudin-1 expression levels in enteroids isolated from jejunal crypts in porcine and mice [109,110,111]. When jejunal explants from weaning piglets were exposed to DON, decreased jejunal scores and MAPK activation were recorded [72]. FB exposure to an ex vivo rat large intestine induced lipid peroxidation, which could alter cell membrane permeability [129]. Apical villi necrosis was found in pig jejunal explant exposure to PAT [123]. DON appears to be the most studied mycotoxin in ex vivo studies. Therefore, it could be speculated that an ex vivo model is also highly applicable for other mycotoxins.

#### 4.1.3. Effects of Mycotoxins on the In Vivo Physical Barrier 

AFB1 and AFM1 have the ability to evaluate caspase-3 and caspase-9 mRNA expression and reduce the Bcl-2/Bax ratio in mice intestinal tissues, suggesting these mycotoxins induce apoptotic events [100]. When broiler chicks were exposed to AFB1, researchers observed increased lactulose:rhamnose (L:R) ratio and diamine oxidase (DAO) in plasma, and reduced clauin-1 and occludin mRNA expression levels in the mid-jejunum [101,102]. DON-mediated impairment of barrier function in rats and grass carp was associated with the depressed crypt depth ratio and TJ amount in the jejunum, which was associated with Wnt/β-catenin and MLCK signaling pathways [108,110,111,113]. In addition, DON-treated weaned piglets exhibited lower disaccharidase (maltase, sucrase, and lactase) activity, suggesting detrimental effects on gut health [112]. Similarly, the effects caused by ZEN in the post-weaning gilts intestinal physical barrier were partially elucidated by ZEN-induced oxidative stress mechanisms [120]. Similar results were shown in rats challenged with ZEN; TJ proteins in the jejunum exhibited down-regulated mRNA expression [119]. The chronic ingestion of FB1 decreased villi height and occludin expression in the piglet intestine [121]. In addition, T-2 toxin treatment disrupted intestinal histology in turkeys and mice, including the duodenum, jejunum, ileum, and colon [124,125]. 

The adverse effects on intestinal morphology in various in vivo models have been recorded for different mycotoxins such as AFs [99,103,104], OTA [107], and DON [72,114,115,116,117]. In addition, a variety of clinical chemistry analytes such as urine, feces, and blood may also reflect defects in intestinal barrier function in vivo (e.g., fatty acid-binding protein and C-reactive protein levels in serum, and fecal hemoglobin and fecal calprotectin) [130,131]. However, these parameters have not been widely used to evaluate the effect of mycotoxin effects. Therefore, they may provide new insights in future studies.

These observations show that when the intestine is affected by short-term, low-concentration mycotoxins, self-regulating cellular abilities maintain the intestinal physical barrier integrity. However, when such damage is sustained and exceeds the self-regulating capabilities, intestinal epithelial cells and TJ proteins become disrupted, leading to barrier compromise. Similarly, long-term exposure of animals to low mycotoxin doses could also adversely affect intestine health.

### 4.2. Effects of Mycotoxins on the Chemical Barrier

The contribution of mycotoxins to intestinal chemical barrier disruption (e.g., mucins and AMPs) has been extensively conducted across in vitro, ex vivo, and in vivo studies (Table 5). To date, hundreds of AMPs have been identified including LEAP-2A, LEAP-2B, hepcidin, and β-defensin1.

#### 4.2.1. Effects of Mycotoxins on the In Vitro Chemical Barrier 

In the co-culture of Caco-2/HT29-MTX cells, AFM1, OTA, and ZEN down-regulated the protein expression of intestinal mucins MUC2, MUC5AC, and MUC5B [43,57,135,136]. Similar results were also observed for DON where it inhibited MUC1, MUC2, and MUC3 mRNA levels in human goblet cells (HT29-16E cells) [134]. T-2 toxin exposure led to a thinned MUC2 layer in HT-29 cells and reduced MUC2 protein expression in Caco-2 cells [124]. Up-regulated porcine β-defensin 1 and β-defensin 2 mRNA expression was also observed following exposure to DON, ZEN, FB1, and NIV, whereas no significant increases in β-defensin 1 and β-defensin 2 secretion occurred in IPEC-J2 cells [137]. In addition, in co-culture studies with Caco-2/HT29-MTX cells, MUC5AC and MUC5B mRNA expression was decreased by DON and NIV exposure, while their protein levels were increased [136]. These differences in mRNA and protein levels may be partly explained by post-transcriptional or post-translational regulatory mechanisms, or protein degradation pathways. Equally, discrepancies in quantification techniques between studies may have also led to these differences, as protein quantification sensitivity is not as high as transcript measurements [144].

#### 4.2.2. Effects of Mycotoxins on Ex Vivo and In Vivo Chemical Barriers 

Few studies have investigated mycotoxin effects on the intestinal chemical barrier. One study showed that, in jejunal explants from pigs, DON time-dependently inhibited mucin mRNA expression levels [134]. In vivo exposure to a mycotoxin-contaminated diet also regulated mucin and other protein productions. Both AMPs and MUC2 mRNA expression levels were inhibited in the intestine of broiler chickens, mice, and juvenile grass carp upon DON exposure [110,111,138,139,140]. Furthermore, reduced goblet cells in the small intestine were observed in OTA and DON treatment [107,112,115,121,140,141]. In contrast, no changes in goblet cells and crude mucin production were observed in laying hens and gilts exposed to AFB1, DON, or ZEN [133,142]. Furthermore, goblet cell hyperplasia (increased goblet cell number) was observed in AF/FB1-challenged broilers and mice exposed to combined DON and ZEN [104,121,132,145]. In general, goblet cell hyperplasia led to increased mucins secretion. T-2 toxin exposure resulted in increased MUC2 mRNA levels in the jejunum of chickens [143]. These observations suggest that the continuous hyper-secretion of mucins is likely to deplete goblet cell number, ultimately disrupting the mucus layer [146].

A variety of responses (increase, no change, and/or decrease) were observed in mycotoxin-induced mucins and goblet cells. Different experimental models using different species could account for these inconsistencies. Precise mechanisms remain unclear, however, and more studies are required to resolve these issues. 

### 4.3. Effects of Mycotoxins on the Immunological Barrier 

Mycotoxin effects on the intestinal immunological barrier in in vitro and in vivo models are summarized (Table 6). 

#### 4.3.1. Effects of Mycotoxins on the In Vitro Immunological Barrier 

OTA down-regulated the gene expression of cyclooxygenase-2 (COX-2) and lipoxygenase-5 (5-LOX) genes in Caco-2 cells, which have been regarded as inflammatory mediators [105]. The mRNA expressions of pro-inflammatory cytokines including interleukin-1 beta (IL-1β), IL-6, IL-8, COX-2, and tumor necrosis factor-alpha (TNF-α) were increased in the DON-exposed porcine intestinal epithelial cells, with DON-mediated inflammation partially dependent on ROS production and MAPK pathway activation [106,151,152,153,154]. Another study also reported that ZEN exerted no effects, or only marginal ones, on cytokine gene expression in IPEC-1 cells [156]. However, other studies reported an increasing tendency towards inflammatory cytokine gene expression and synthesis in ZEN-exposed IPEC-1 cells [91,118,157]. In addition, the ZEN metabolites α-ZOL and β-ZOL decreased pro- and anti-inflammatory cytokine expressions [118]. The exposure of FB1, in a concentration-dependent manner, to IPEC-1 cells depressed IL-8 mRNA and protein expression, while no changes in IL-8 secretion were recorded in HT-29 cells [93,158]. These studies show that mycotoxins, especially ZEN, exert different inflammatory cytokine expression profiles, suggesting underlying mechanisms are complex and require further in-depth study. Additionally, there is no evidence on the impact of AFs on intestinal inflammation. Therefore, more studies are required to fill this important knowledge gap.

#### 4.3.2. Effects of Mycotoxins on the In Vivo Immunological Barrier 

AFB1 and AFM1 down-regulated pro-inflammatory cytokine production in the small intestine of mice and pigs [100,147]. AFB1 also affected sIgA and polymeric immunoglobulin receptor (pIgR) expression, thereby affecting sIgA transport in epithelial cells into mucus layers. Transcriptome analysis revealed that seven processes or pathways related to the immune system were grouped in *Litopenaeus vannamei* as the result of AFB1 challenge [148]. OTA exposure resulted in up-regulated IL-1β and TNF-α mRNA expression in the small intestine of broiler chickens and pekin ducklings [107,150]. Similar results were also observed for DON and FB1 exposure, where cytokine production was significantly up-regulated in the intestine of piglets and juvenile grass carp [121,138,155]. In other studies, in the intestines of pregnant dams, weaned pups, and pigs treated with ZEN and FB1, pro-inflammatory cytokine mRNA expression was significantly decreased [119,158].

CD4^+^ cells are involved in cytokine production, and CD8^+^ cells play a crucial role in protecting host target cells [160,161]. Mycotoxin-contaminated diets led to decreased CD4^+^ and CD8^+^ cell percentages in chickens [162]. In addition, CD4^+^ cell numbers were decreased in the jejunum of broilers treated with AFB1 [149]. In addition, CTN administration altered immune cell (CD4^+^, CD8^+^, and CD19^+^) populations in the small intestine of mice [159]. Similarly, cell morphology effects have been observed in these models; in AF-challenged broiler chicks, the lamina propria lymphoid follicle diameter (LLFD) and lamina propria lymphoid follicle number (LLFN) in the jejunum were increased [104]. In addition, DON exposure increased the lymphocyte number in the intestine of pigs fed DON-contaminated diets [141,142], indicating mycotoxin-mediated inflammatory responses.

These data reflect the limited research on disrupted intestinal chemical barriers induced by mycotoxins in ex vivo models. While these studies have enlightened the literature on AFs, OTA, DON, and ZEN, the effects of other mycotoxins such as FB1 and PAT require greater attention.

### 4.4. Effects of Mycotoxins on the Microbial Barrier

Several studies have shown the impairment to the intestinal microbial barrier is caused by mycotoxins (Table 7). 

#### 4.4.1. Effects of Mycotoxins on the In Vitro Microbial Barrier 

The simulator of the human intestinal microbial ecosystem (SHIME) was used to investigate OTA effects, and it showed that gut microbiota diversity was altered, with the loss of beneficial species *Lactobacillus reuteri* [167]. In another study, *Escherichia coli* (*E. coli)* were translocated across IPEC-J2 cell monolayers when induced by DON [152]. This finding was consistent with previous studies, demonstrating that bacterial translocations across IPEC-1 and Caco-2 cell monolayers are induced by DON, NIV, and PAT [166,169]. Importantly, bacterial translocation appears to be concentration- and species-dependent [179], with incubation times and cell line types potentially affecting the results [166]. 

#### 4.4.2. Effects of Mycotoxins on the In Vivo Microbial Barrier 

In broiler chicks, the dietary supplementation of AFs resulted in markedly increased ileal bacterial counts of *E. coli, Klebsiella, Salmonella,* and total negative bacteria [104,132,165]. In addition, AFB1 reduced intestinal bacterial flora diversity in rats, shrimp, and mice [163,164,180]. In AFB1-treated shrimp, the application of high-throughput sequencing analysis showed that *Bacteroidetes* relative abundance decreased, and the abundance of *Proteobacteria* and *Firmicutes* increased [180]. These results were consistent with a previous study demonstrating that OTA exposure significantly increased the relative abundance of *Lactobacillus*, but decreased overall gut microbiota diversity in rats [168].

Cecal species’ richness and evenness were decreased upon DON exposure in broiler chickens and mice, *Bacteroidetes* and *Firmicutes* abundance increased, while *Proteobacteria* decreased, suggesting an overall impaired gut microbiota community [170,171]. A fecal microbiota analysis explored DON-induced gut microbiota changes in nude mice and pigs and showed that DON treatment generated a higher abundance of *Clostridiales*, *Lachnospiraceae,* and fecal aerobic mesophilic bacteria [140,172]. In addition, DON exposure increased *Bacteroides* and *Prevotella* genera levels but decreased *E. coli* levels in a model of human microbiota-associated rats [174]. 

The biolog EcoPlate method showed that ZEN exposure significantly reduced *E. coli*, *Clostridium perfringens*, and *Enterobacteriaceae* levels in gilts [177]. Additionally, ZEN increased *Desulfovibrio* and decreased *Lactobacillus* in mouse colon tissue [176]. After exposure to FBs, the digestive microbiota balance was impaired in pigs [178]. In contrast, AFB1 and DON exerted no effects on gut microbiota diversity and relative abundance in broilers and rats [102,173,175].

In general, microbiota communities in the colon, cecum, and feces are representative of the intestinal bacterial flora. However, microbiota differences in different intestinal sections are still unclear. Similarity, studies on mycotoxin (i.e., ZEN and FB1) mediated alterations in the intestinal microbial barrier are limited. Therefore, more research is required in this area.

## 5. Contribution of a Leaky Gut to Intestinal Inflammatory Disease

Currently, links between mycotoxin exposure and some human carcinogenic and teratogenic diseases, including Reye’ syndrome, cirrhosis, hepatitis, and esophagus cancer, have been demonstrated [181]. However, potential links between mycotoxins and human chronic intestinal inflammatory diseases remain unclear. Among the chronic intestinal inflammatory diseases, inflammatory bowel diseases (IBDs) including ulcerative colitis (UC), Crohn’s disease (CD), and celiac disease (CeD) are well studied. The dysfunction of intestinal barrier and increased intestinal permeability, also known as “leaky gut”, are believed to play a prominent role in the etiology of these diseases [182,183]. 

Intestinal epithelial cell death and abnormal TJ expression cause increased transcellular and paracellular transport. These perturbations may cause increased access of different molecular weight xenobiotics and bacterial translocation, ultimately activating local and systemic immune responses [23,184]. Among the abnormalities of TJs, claudin switching is the most notable, which refers to the balance between different members of the claudin family [185,186]. In particular, the down-regulation of several tight claudins, such as claudins 3, 4, 5, 7, and 8, and the increased expression of leaky claudins, such as claudin 2 and 15, are often associated with an inflamed gut, including UC, CD, CeD, and irritable bowel syndrome [185,187,188,189,190]. In addition, reduced TJ expression could accelerate mucosal inflammation (ulceration and colitis), as observed in claudin 2 and claudin 7 knockout mice, and claudin 2 overexpression investigations in mice [191,192,193].

In turn, inflammatory mediators themselves could also generate negative effects on the TJs. Thus, inflammatory responses caused by the initial increases in intestinal permeability could be responsible for a stronger disruptive effect in the intestinal physical barrier [194]. Indeed, several cytokines (e.g., IL-1β, IL-6, IL-9, IL-17, IL-23, IFN-γ, and TNF-α) have been shown to alter TJ expression in in vitro and in vivo models [195,196,197,198,199,200,201,202,203].

In addition to intestinal physical and immunological barriers, chemical barrier disruption also leads to inflammatory disease. Perturbations in intestinal mucus layers, including mucus composition and production, contribute to human chronic intestinal inflammatory disease. In addition, significant modifications to gut microbiota number and composition were observed in IBD patients and were characterized by decreased anaerobic bacteria and increased Enterobacteria numbers [181]. The intestinal microflora may also affect mucus production and TJ expression, potentially leading to inflammatory responses [204,205]. As such, the intestinal barrier is an interconnected system. Therefore, disruption to one or more components may result in human intestinal disease [20,206]. Several factors are involved in the induction and persistence of the chronic intestinal inflammatory diseases: (i) abnormal TJs composition, (ii) altered production of mucus layers, (iii) changes in intestinal immunity, and (iv) modification of intestinal microflora or increased bacterial translocation (Figure 2). Given that healthy relatives of CD patients manifested increased intestinal permeability without disease [207,208], we conclude that a compromised intestinal barrier is not the sole factor causing inflammatory diseases, but intestinal dysfunction could exacerbate inflammation and enhance its severity [21,183].

## 6. Interactive Effects of Mycotoxins on Intestinal Barrier

Most fungi produce one or more mycotoxins. Therefore, the co-occurrence of these molecules in disease should receive more attention. A previous study demonstrated that approximately 28 mycotoxins were quantified by a liquid chromatography tandem mass spectrometry (LC-MS/MS) multi-toxin method, in samples collected from Burkina Faso and Mozambique [209]. A recent study also showed a greater co-exposure phenomenon, where the analyzed maize samples were contaminated by 5 to 41 mycotoxins in Malawi [210]. In addition, mycotoxin co-occurrence was also identified in oil, dairy milk, and infant formulas [211,212,213]. After ingestion of mycotoxin-contaminated food, these molecules may be absorbed into the blood. To corroborate this, UHPLC-MS/MS analysis identified 26 mycotoxin biomarkers, including AFB1, OTA, FB1, ZEN, and DON, in plasma and urine samples from 260 rural residents in China [214]. Equally, similar data have been observed in different populations across the globe [215,216,217,218]. 

The interactive effects of multi-mycotoxins are classified as synergistic, additive, and antagonistic, and they represent the effects of a components’ mixture are higher, equal, and lower than the sum of the effects induced by individual components. For the intestinal physical barrier, AFM1 cytotoxicity against Caco-2 cells was enhanced in the presence of OTA, ZEA, and α-ZOL, with additive and synergistic effects demonstrated by most combinations [219]. The additive and synergistic effects on reduced cell viability were demonstrated by combining DON and ZEN in a bi-culture Caco-2, THP-1, and HepaRG cell system [220]. The effects on increased intestinal permeability and decreased TJ expression in differentiated Caco-2 cells exposed to a combination of AFM1 and OTA were significantly stronger than the individual mycotoxins, performing additive and synergistic effect [54]. In addition, non-toxic OTA aggravated DON-induced TJ disruption in IPEC-J2 cells, suggesting a synergistic effect was at play [106]. 

From a chemical barrier perspective, an AFM1 and OTA combinatorial approach produced a synergistic effect by depressing mucin expression in Caco-2/HT29-MTX co-cultures [43]. In contrast, antagonistic interaction of combined DON and ZEN was observed for MUC5AC expression levels [135]. Non-additive (including synergistic and antagonistic) effects were also shown for the modulation of mucins and β-defensins expression, induced by combinations of DON, FB1, ZEN, and NIV [136,137]. Furthermore, different interactions were observed in the combination of DON and FB on intestinal immunological changes [121]. Additionally, the antagonist effect was observed in the combined DON and ZEN on gut microbiota levels [177]. 

To protect the gut from mycotoxin assault, several approaches have been undertaken to assess the protective effects of different materials on intestinal dysfunction. Lactoferrin [79], grape seed [147], silymarin [132], yeast cell wall [102], *Lactobacillus plantarum* [100], and mannanoligosaccharides [104] have alleviated AF-induced intestinal disturbances in human intestinal epithelial cells, pigs, broiler chicks, and mice. Probiotic *Bacillus subtilis* [82], curcumin [150], and selenium-rich yeast [107] have attenuated OTA-induced disruption of intestinal barrier in Caco-2 cells, ducks, and broiler chickens. Similarly, mixtures of *Lactobacilli* [156] and *Bacillus cereus* BC7 strain [180] efficiently renormalized ZEN-induced perturbations of intestinal inflammatory responses and microbiota. Resveratrol [152], probiotic *Lactobacillus rhamnosus* GG [140], methionine and its hydroxyl analogues [110], and hydrolyzed wheat gluten [111] all reduced DON-induced intestinal damage. Eugenol also protected against CTN-induced intestinal cytotoxicity in HCT116 cells [9]. More importantly, *L. rhamnosus* GG was reported to improve intestinal barrier functions in mice exposed to DON and ZEN combination [145].

## 7. Conclusions

Mycotoxin contamination of human food and animal feed results in economic loss and health detriments. In addition to the carcinogenic, mutagenic, and genotoxic effects mediated by these molecules, mycotoxin-induced impairment of the intestinal barrier has also gained considerable attention in recent years. To assess intestinal toxicity, in vitro epithelial barrier investigations frequently use human (Caco-2 and HT29 cells) and porcine epithelial cells (IPEC-1and IPEC-J2) as well as enteroids. Ex vivo models are represented by explants from mice and pigs. The predominant in vivo models use rodents (mice and rats) and pigs, followed by chickens and fish. Based on these varied models, our review revealed that the mycotoxins, i.e., AFs, OTA, ZEN, DON and FB1, decreased TJ protein expression levels, depressed mucins and AMPs secretion, activated cytokine production, and altered gut microflora composition. These elements represent the intestinal physical, chemical, immunological, and microbial barrier targeted by mycotoxins. 

It is entirely plausible that mycotoxins are implicated in human chronic intestinal inflammatory disease. In this review, we showed evidence linking a compromised intestinal barrier to inflammatory disease, but further studies are required to fully confirm this hypothesis. Therefore, underlying mechanisms remain an issue. More studies are required to investigate how to control and decrease mycotoxin-contaminated foods and animal feeds, to protect human and animal health, and to reduce economic loss.

## Figures and Tables

**Figure 1 toxins-12-00619-f001:**
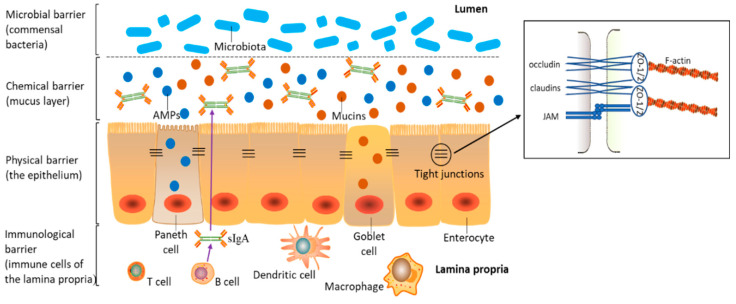
Normal intestinal homeostasis. The intestinal barrier is equipped with four levels to protect the intestine from external stimuli. This includes a physical barrier (a single layer of semi-permeable epithelial cells), chemical barrier (a mucus layer consisting of mucins and antimicrobial peptides, secreted by goblet cells and Paneth cells, respectively), immunological barrier (immune cells in the lamina propria and secreted immune mediators such as cytokines and secretory immunoglobulin A (sIgA)), and microbial barrier (commensal bacteria in the intestinal lumen). Adjacent epithelial cells are connected by tight junctions, which are composed of transmembrane proteins, junctional adhesion molecules (JAMs), claudins, and occludin that are linked to the actin cytoskeleton through zonula occludens (ZO) proteins.

**Figure 2 toxins-12-00619-f002:**
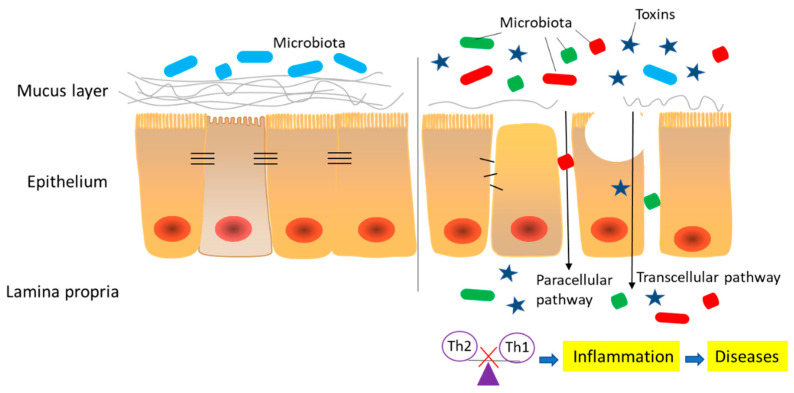
Summary of the negative effects induced by mycotoxins on intestinal barrier. Relevant aspects include (i) increased permeability (paracellular and transcellular transport), which is induced by disrupted epithelial cells and tight junctions, and (ii) the thinned mucus layer. The compromised intestinal barrier results in the penetration of xenobiotics of different molecular weights and bacterial translocation, ultimately contributing to an imbalance of inflammatory responses and the activation of local and systemic immunity, causing the occurrence of inflammatory-related diseases.

**Table 1 toxins-12-00619-t001:** The overview of mycotoxins.

Mycotoxin	Toxic Effects	IARC Classification	Health Guidance Value	References
AFB1, AFM1	carcinogenic	Group 1	As low as reasonably achievable (ALARA principle)	[5,6]
OTA	nephrotoxic, teratogenic, immunotoxic, neurotoxic	Group 2B	TWI = 120 ng/kg bw/w PTWI = 100 ng/kg bw/w	[5,6]
FB1	hepatotoxicity, nephrotoxicity	Group 2B	PMTDI (FB1+FB2 +FB 3) = 2 μg/kg bw/d TDI (FB1+FB2 +FB3) = 2 μg/kg bw/d	[1,5,6]
DON	immunotoxic	Group 3	TDI = 1 μg/kg bw/d	[5,6,7,8]
ZEN	reproductive toxicity	Group 3	PMTDI = 0.5 μg/kg bw/d TDI = 0.25 μg/kg bw/d	[5,6]
PAT	hepatotoxicity	Group 3	PMTDI = 0.4 μg/kg bw/d	[1,5,9]
NIV	immunotoxicity, hematotoxicity, myelotoxicity	Group 3	TDI = 1.2 μg/kg bw/d	[2,5,10]
T-2, HT-2	toxic on the skin and mucous membranes	Group 3	PMTDI (T-2+HT-2) = 0.06 μg/kg bw/d TDI (T-2+HT-2) = 0.1 μg/kg bw/d	[5]

Group 1, carcinogenic to humans; Group 2A, probably carcinogenic to humans; Group 2B, possibly carcinogenic to humans; Group 3, not classifiable as to its carcinogenicity to humans.

**Table 2 toxins-12-00619-t002:** The characteristics of in vitro, ex vivo, and in vivo intestine models.

Models	Types	Advantages	Limitations	References
In vitro-2D intestinal model	Caco-2 cells, IPEC-1 cells, IPEC-J2 cells, IPI-2I cells and PSI-1 cells, co-culture of different cell lines)	well-established and relatively cheap	only containing a single cell type without villus and crypt domain	[39,40,41,42,43,44]
In vitro-3D intestinal model	enteroids, also known as organoids or mini-guts	partially recapitulate the anatomy of native epithelium, have the ability to passage at an almost unlimited scale	the effects of substances on the luminal side are poorly investigated, considerable cost, do not contain the immune and stromal cells	[41,45,46,47,48,49]
Ex vivo	applied in humans, rodents, swine, poultry and horse	a more accurate model to mimic the physiology in vivo	fail to achieve long-term culture, careful and laborious preparation	[45,50,51]
In vivo	commonly used models include mouse, rat, chicken, turkey, fish, pig, sheep and bovine	provide the information based on the whole animals, thus they could corroborate the toxicity in humans effectively	the use of live animals should follow 3R (replacement, reduction and refinement) principle	[1]

**Table 3 toxins-12-00619-t003:** Modulation of intestinal epithelial cells (physical barrier) induced by mycotoxins.

Model	Dose/Administration Route	Exposure Time	Technique	Damage	References
Aflatoxin					
Caco-2 cells	AFB1: 1–50 μM	24 h	Alamar blue assay	Decrease cell viability	[77]
Caco-2 cells	AFB1: 0.01–1 μg/mL AFM1: 0.01–1 μg/mL	24, 48, 72 h	MTT assay	AFB1: decrease cell viability AFM1: decrease cell viability	[78]
Caco-2 cells	AFB1: 4 μg/mL AFM1: 4 μg/mL	24 h	MTT assay	AFB1: decrease cell viability AFM1: decrease cell viability	[79]
Caco-2 cells	AFM1: 0.0005–4 μg/mL	48 h	RNA-Seq, CCK-8 assay, Flow cytometry analysis	No effect on cell viability Induce cell cycle arrest	[80]
Caco-2/HT29-MTX cells	AFM1: 0.05, 4 μg/mL	48 h	CCK-8 assay	Decrease cell viability	[43]
Ochratoxin A					
Caco-2/HT29-MTX cells	OTA: 0.05, 4 μg/mL	48 h	CCK-8 assay	Decrease cell viability	[43]
IPEC-J2 Cells	OTA: 0.5–32 μM	6, 12, 24 h	MTT assay	Decrease cell viability	[81]
Caco-2 cells	OTA: 0.1–30 μM	24 h	RNA-Seq, CCK-8 assay, Flow cytometry analysis	Decrease cell viability Arrest cell cycle in G2/M phase Induce apoptosis	[82]
Caco-2 cells	OTA: 0.0005–4 μg/mL	48 h	RNA-Seq, Flow cytometry analysis	Induce cell apoptosis	[83]
Caco-2 cells	OTA: 0.5–160 μM	48 h	MTS assay	Decrease cell viability	[84]
Deoxynivalenol					
Caco-2 cells	DON: 0.25–30 μM	48 h	MTS assay	Decrease cell viability	[84]
HT-29 cells	DON: 125–2000 ng/mL	24 h	Western blot analysis	Induce cell apoptosis	[24]
IEC-6 cells	DON: 0.5–80 μM	24 h	Propidium iodide staining	Induce cell apoptosis	[85]
IPEC-J2 Cells	DON: 200, 2000 ng/mL	24, 48, 72 h	BrdU incorporation assay, Western blot analysis	Decrease cell viability Induce cell apoptosis	[86]
IPEC-1 cells, IPEC-J2 Cells	DON: 100–4000 ng/mL	24, 48, 72 h	MTT assay	Decrease cell viability	[87]
IPEC-J2 Cells	DON: 1–20 μg/mL	72 h	Flow cytometry analysis	Decrease cell viability Induce cell apoptosis	[88]
Zearalenone					
IPEC-J2 Cells	ZEN: 40 μM	24 h	CCK-8 assay, Flow cytometry analysis, RNA-Seq	Decrease cell viability Arrest cell cycle in the G2/M phase	[89]
IPEC-J2 cells	ZEN: 6, 8 μg/mL	12–48 h	MTT assay, PCR	Decrease cell viability Induce cell apoptosis	[90]
IPEC-1 cells	ZEN: 0.1–100 μM	24 h	XTT assay, Microarray assay	Decrease cell viability	[91]
HCT116 cells	ZEN: 0–320 μM	48 h	Methylene blue staining assay	Increase cell viability at very low concentrations, decrease cell viability at high concentrations	[92]
Fumonisin B1					
HT-29 cells	FB1: 1.1–69 μM	72 h	MTT assay	Decrease cell viability	[93]
Patulin					
Caco-2 cells	PAT: 1–150 μM	24 h	MTT assay	Decrease cell viability	[94]
Caco-2 cells	PAT: 0.7–18 μM	24 h	MTT assay	Decrease cell viability	[95]
HCT116 cells	PAT: 5–25 μM	24 h	FDA assay, Western blot analysis	Decrease cell viability Induce cell apoptosis	[96]
HCT116 cells	PAT: 1.25–20 μM	1–4 days, 24 h	MTT assay, Western blot analysis	Decrease cell viability Induce apoptotic cells death	[97]
Citrinin					
HCT116 cells	CTN: 150 μM	24 h	MTT assay	Decrease cell viability	[98]

**Table 4 toxins-12-00619-t004:** Modulation of tight junctions (physical barrier) induced by mycotoxins.

Model	Dose/Administration Route	Exposure Time	Technique	Damage	References
Aflatoxin					
Caco-2 cells	AFM1: 0.12, 12 μM	48 h	Western blot analysis, Immunofluorescent staining	Decrease in TEER value Increase in permeability of LY and 4 and 40 kDa FITC-dextran Decrease the protein expression of ZO-1, occludin, claudin-4, and claudin-3 Affect the distribution pattern of ZO-1, occludin, claudin-4, and claudin-3	[54]
Caco-2/HT29-MTX cells	AFM1: 12 μM	48 h	Transmission electron micrographs	Affect the distribution pattern of ZO-1, occludin, claudin-4, and claudin-3 disrupt TJs structure	[57]
Rat (Wistar, *n* = 35)	AFB1: 2.5 mg/kg Intraperitoneal administration	7 days	Histopathological analysis	Villi degeneration of duodenum and ileum	[99]
Mice (Balb/c, *n* = 60)	AFB1: 100 μg/kg b.w. AFM1: 100 μg/kg b.w. Oral administration	14 days	RT-PCR, Western blot analysis	Induce small intestine apoptosis	[100]
Broiler chicks (Ross 708, *n* = 288)	AFB1: 1.5 mg/kg Contaminated feed	20 days	Serum biochemistry, RT-PCR	Increase in serum lactulose/rhamnose ratio Increase in transcript level of claudin-1 in jejunum	[101]
Broiler (Cobb, *n* = 576)	AFB1: 40 μg/kg Contaminated feed	21 days	Serum biochemistry, RT-PCR	Increase in serum diamine oxidase concentration Decrease in mRNA level of occludin and claudin-1 in jejunum	[102]
Duck (Cherry Valley, *n* = 640)	AFB1: 195.4 μg/kg Contaminated feed	35 days	Intestinal morphology analysis	Increase in crypt depth, villus width of duodenum Increase in villus height, villus width of jejunum	[103]
Broiler chicks (Ross 308, *n* = 336)	AFs 0.5 and 2 mg/kg feed Contaminated feed	28, 42 days	Intestinal morphology analysis	Decrease in villi height to crypt depth ratio	[104]
Ochratoxin A					
Caco-2 cells	OTA: 0.12, 12 μM	48 h	Western blot analysis, Immunofluorescent staining	Decrease in TEER value Increase in permeability of LY and 4 and 40 kDa FITC-dextran Decrease in protein expression of ZO-1, occluding, claudin-4, and claudin-3 Affect the distribution pattern of ZO-1, occluding, claudin-4, and claudin-3	[54]
Caco-2 cells	OTA: 5–45 μM	3, 12, 24 h	TEER measurement	Decrease in TEER value	[105]
Caco-2 cells	OTA: 15 μM	5 h	Transmission electron microscope, Immunofluorescent staining	Reduce the microvilli on cell surface Alter the localization and distribution of claudin-1 and ZO-1	[82]
IPEC-J2 cells	OTA: 4–128 μM	6, 12, 24 h	Measurement of epithelial monolayer paracellular permeability	Decrease in TEER value Increase in permeability of 4 kDa FITC-dextran	[106]
Broiler chickens (*n* = 80)	OTA: 50 μg/kg b.w. Oral administration	21 days	Intestinal morphology analysis	Decrease in villi height to crypt depth in duodenum, jejunum, and ileum	[107]
Deoxynivalenol					
Caco-2 cells	DON: 1–30 μM	24, 40, 48 h	Western blot analysis, Immunofluorescent staining	Decrease in TEER value Increase in permeability of 4 kDa FITC-dextran Decrease in protein expression of claudin-7, occludin and E-cadherin Alter the localization and distribution of occludin and E-cadherin	[108]
IPEC-J2 cells	DON: 250, 500 ng/mL	0.5–120 h	TEER measurement	Decrease in TEER value	[109]
IPEC-J2 cells	DON: 0.5–16 μM	6, 12, 24 h	Western blot analysis, Immunofluorescent staining	Decrease in TEER value Increase in permeability of 4 kDa FITC-dextran Decrease in protein expression of claudin-3, and claudin-4 Alter the localization and distribution of claudin-3, and claudin-4	[106]
Mouse enteroids, Mice (C57BL/6, *n* = 72)	DON: 250 ng/mL, DON: 2 mg/kg b.w. Oral administration	72 h, 1–12 days	Immunofluorescent staining, Western blot analysis	Alter the localization and distribution of claudin-1 Decrease in protein expression of claudin-1, and ZO-1 in jejunum	[110]
Piglets jejunal explants	DON: 5, 10 μM	4 h	histological analysis	Induce histological lesions on the intestine	[72]
Mice (C57BL/6, *n* = 72)	DON: 2 mg/kg b.w. Oral administration	14 days	Serum biochemistry, Intestinal morphology analysis	Increase in serum diamine oxidase activity Decrease in villus/crypt ratio	[111]
Rat (Wistar, *n* = 32)	DON: 8.2 mg/kg feed Contaminated feed	28 days	Histological and morphometric assessment, Immunohistochemical assessment	Decrease in crypt depth in jejunum, Decrease in the expression of occludin and E-cadherin in jejunum	[108]
Pig (piglet, *n* = 24)	DON: 50 μg/kg b.w. Contaminated feed	15 days	Disaccharidases activity	Decrease in maltase, sucrase and lactase activity of the small intestine (duodenum, proximal and medium jejunum and ileum)	[112]
Fish (juvenile grass carp, *n* = 1440)	DON: 27–1515 μg/kg diet Contaminated feed	60 days	RT-PCR	Decrease in the mRNA levels of ZO-1, ZO-2b, occludin, claudin-c, -f, -7a, -7b, -11 in fish intestine, Increase in the mRNA levels of claudin-12, -15a in fish intestine	[113]
Broiler chickens (*n* = 40)	DON: 10 mg/kg feed Contaminated feed	35 days	Thiobarbituric acid reactive substance estimation	Increase in thiobarbituric acid reactive substance level, an indicator of oxidative stress, in jejunum	[114]
Pig (piglet, *n* = 20)	DON: 2 mg/kg feed Contaminated feed	28 days	Histological assessment	Increase in the lesional score in intestine	[115]
Pig (piglet, *n* = 48)	DON: 1000–3000 μg/kg feed Contaminated feed	21 days	Histological assessment	Decrease in villi height/crypt depth ratio in jejunum	[116]
Pig (piglet, *n* = 12)	DON: 2.3 mg/kg feed Contaminated feed	20 days	Histological analysis	Decrease in the histological score in the jejunum	[72]
Broiler (Ross, *n* = 75)	DON: 1.7, 12.2 mg/kg feed Contaminated feed	35 days	Histological analysis	Decrease in relative density (weight: length) of the small intestine	[117]
Zearalenone					
IPEC-1 cells	ZEN: 25, 50 μM α-ZOL: 25, 50 μM β-ZOL: 25, 50 μM	1–10 days	TEER measurement	ZEN: no effect in TEER value α-ZOL: decrease in TEER value β-ZOL: decrease in TEER value	[118]
Rat (Sprague-Dawley, *n* = 96)	ZEN: 1.3–146.0 mg/kg Contaminated feed	7 days	RT-PCR	Decrease in the mRNA expression of claudin-4 and occludin in jejunum	[119]
Pig (gilt, *n* = 40)	ZEN: 0.5–1.5 mg/kg Contaminated feed	10 days	RT-PCR, Western blot analysis	Increase in the expression of oxidative stress related proteins	[120]
Fumonisin B1					
IPEC-J2 cells	FB1: 50, 100 μM HFB1: 50, 100 μM	1-9 days	TEER measurement	FB1: decrease in TEER value HFB1: decrease in TEER value	[42]
Pig (piglet, *n* = 24)	FB1: 6 mg/kg Contaminated feed	35 days	Western blot analysis	Decrease in the protein expression of occludin in ileum	[121]
Patulin					
Caco-2 cells	PAT: 3–50 μM	24 h	Western blot analysis	Decrease in TEER value Decrease in protein expression of ZO-1	[94]
Caco-2 cells	PAT: 5–100 μM	24 h	TEER measurement	Decrease in TEER value	[95]
Caco-2 cells	PAT: 50 μM	3–72 h	Western blot analysis, Immunofluorescent staining	Decrease in TEER value Alter the localization and distribution of claudin-4, occludin and ZO-1 Decrease in protein expression of ZO-1	[122]
Pigs jejunal explants	PAT: 10–100 μM	4 h	Histological and morphometric analysis	Induce apical villi necrosis and alter lateral intercellular disruption	[123]
*T-2 toxin*					
Caco-2 cells	T-2: 50–100 ng/mL	24 h	Western blot analysis	Decrease in TEER value Decrease in the expression of occludin	[124]
Turkey poults (*n* = 24)	T-2: 241–982 ppb Contaminated feed	32 days	Histological analysis	Decrease in villi height in the jejunum	[125]
Mice (BALB/c, *n* = 30)	T-2: 0.5, 2.0 mg/kg b.w. Oral administration	28 days	Histological analysis	Decrease in villi height in the ileum	[124]

**Table 5 toxins-12-00619-t005:** Modulation of the intestinal chemical barrier induced by mycotoxins.

Model	Dose/Administration Route	Exposure Time	Technique	Damage	References
Aflatoxin					
Cao-2/HT29-MTX cells	AFM1: 0.05, 4 μg/mL	48 h	RT-PCR, ELISA	Change the mRNA and protein expression level of MUC2, MUC5AC and MUC 5B in different proportions of co-cultured cells	[43]
Cao-2/HT29-MTX cells	AFM1: 12 μM	48 h	RT-PCR, ELISA	No effect on the mRNA and protein expression level of MUC2, MUC5AC and MUC 5B	[57]
Broiler chicks (Ross 308, *n* = 336)	AFs (AFB1+AFB2+AFG1+AFG2) 0.5 and 2 mg/kg feed Contaminated feed	28, 42 days	Histological analysis	Increase in the goblet cell counts at 28 and 42 d	[104]
Broiler chicks (Ross 308, *n* = 336)	AFs (AFB1+AFB2+AFG1+AFG2) 0.5 and 2 ppm feed Contaminated feed	28, 42 days	Histological analysis	Increase in the goblet cell counts at 28 and 42 d	[132]
Hens (Hyline W36, *n* = 64)	AFB1: 0.5–2.0 mg/kg	14 days	Histological analysis	No changes in goblet cell number and crude mucin production	[133]
*Ochratoxin A*					
Cao-2/HT29-MTX cells	OTA: 0.05, 4 μg/mL	48 h	RT-PCR, ELISA	Modulate the mRNA level of MUC2, MUC5AC and MUC 5B, Increase in the protein expression of MUC2 and MUC5B at low concentration, while decrease at high concentration	[43]
Broiler chickens (*n* = 80)	OTA: 50 μg/kg b.w. Oral administration	21 days	Histological analysis	Decrease in goblet cells number in the small intestine	[107]
*Deoxynivalenol*					
HT29-16E cells	DON: 0.1–100 μM	3–48 h	RT-PCR	Decrease in the transcript level of MUC1, MUC2 and MUC3	[134]
Cao-2 cells	DON: 2 μM	5 min–24 h	RT-PCR	Increase in the transcript level of MUC5AC	[135]
Cao-2/HT29-MTX cells	DON: 2 μM	48 h	RT-PCR, ELISA	Decrease in the transcript level of MUC5AC and MUC5B in the 90:10 ratio	[136]
IPEC-J2 cells	DON: 2 μM	48 h	RT-PCR, ELISA	Increase in the transcript level of β-defensin1 and β-defensin2, while no effect on protein expression	[137]
porcine intestinal explants	DON: 10 μM	8, 12 h	RT-PCR	Decrease in the transcript level of MUC1, MUC2 and MUC3	[134]
Fish (juvenile grass carp, *n* = 1440)	DON: 318–1515 μg/kg diet Contaminated feed	60 days	RT-PCR	Decrease in the mRNA expression of MUC2 and AMPs (β-defensin1, hepcidin, LEAP-2A and LEAP-2B) in proximal, middle and distal intestine	[138]
Broiler Chickens (Ross 308, *n* = 112)	DON: 4.6 mg/kg feed Contaminated feed	15 days	RT-PCR	Decrease in the mRNA expression of MUC2 in duodenum	[139]
Mice (BALB/c, *n* = 42)	DON: 3.0 mg/kg Gavage	15 days	RT-PCR	Decrease in the mRNA expression of MUC2	[140]
Mice (C57BL/6, *n* = 72)	DON: 2 mg/kg b.w. Oral administration	1–12 days	Immunohistochemistry staining	Decrease in the MUC2^+^ cells and LYZ^+^ cells number in jejunum	[110]
Mice (C57BL/6, *n* = 72)	DON: 2 mg/kg b.w. Oral administration	14 days	Immunohistochemistry staining	Decrease in the MUC2^+^ cells and LYZ^+^ cells number in jejunum	[111]
Pig (piglet, *n* = 24)	DON: 3.0 mg/kg Contaminated feed	35 days	Histological analysis	Decrease in the goblet cells number in jejunum and ileum	[121]
Pig (piglet, *n* = 20)	DON: 1.5 mg/kg Contaminated feed	28 days	Histological analysis	Decrease in the number of goblet cells in jejunum and ileum	[115]
Mice (BALB/c, *n* = 42)	DON: 3.0 mg/kg Gavage	15 days	Histological analysis	Decrease in the goblet cells number	[140]
Pig (piglet, *n* = 24)	DON: 50 μg/kg b.w. Contaminated feed	15 days	Histological analysis	Decrease in the goblet cells number in villi, but no effects at crypts level	[112]
Pig (growing pigs, *n* = 24)	DON: 3, 6, 12 mg/kg feed Contaminated feed	21 days	Histological analysis	Decrease in the goblet cells number of jejunum of pigs fed with diets 6 mg/kg DON-contaminated	[141]
Pig (gilt, *n* = 72)	DON: 12 μg/kg b.w. Contaminated feed	7–42 days	Histological analysis	No effect on the goblet cells number of duodenum	[142]
*Zearalenone*					
Cao-2 cells	ZEN: 40 μM	5 min–24 h	RT-PCR	Decrease in the mRNA expression of MUC5AC	[135]
Cao-2/HT29-MTX cells	ZEN: 40 μM	48 h	RT-PCR, ELISA	Decrease in the transcript level of MUC5AC	[136]
IPEC-J2 cells	ZEN: 40 μM	48 h	RT-PCR, ELISA	Increase in the transcript level of β-defensin1 and β-defensin2, while no effect on protein expression	[137]
Pig (gilt, *n* = 72)	ZEN: 40 μg/kg b.w. Contaminated feed	7–42 days	Histological analysis	No effect on the goblet cells number of duodenum	[142]
Fumonisin B1					
IPEC-J2 cells	FB1: 40 μM	48 h	RT-PCR, ELISA	Increase in the mRNA expression of β-defensin1 and β-defensin2, while no effect on protein expression	[137]
Pig (piglet, *n* = 24)	DON: 6.0 mg/kg Contaminated feed	35 days	Histological analysis	Inductive effect on goblet cell in jejunum	[121]
T-2 toxin					
HT-29 cells Cao-2 cells	T-2: 50–100 ng/mL	24 h	Immunofluorescence staining, Western blot analysis	Decrease in the mucus layer in Caco-2 cells and HT-29 cells Decrease the protein expression of MUC2	[124]
Chickens (*n* = 20)	T-2: 145 μg/kg diet Contaminated feed	14 days	RT-PCR	Increase MUC2 mRNA expression level in jejunum	[143]
Nivalenol					
Cao-2/HT29-MTX cells	NIV: 2 μM	48 h	RT-PCR, ELISA	Decrease in the mRNA expression of MUC5AC and MUC5B in the 90:10 ratio	[136]
IPEC-J2 cells	NIV: 2 μM	48 h	RT-PCR, ELISA	Increase in the transcript level of β-defensin1 and β-defensin2, while no effect on protein expression	[137]

**Table 6 toxins-12-00619-t006:** Modulation of the intestinal immunological barrier induced by mycotoxins.

Model	Dose/Administration Route	Exposure Time	Technique	Damage	References
Aflatoxin					
Mice (Balb/c, *n* = 60)	AFB1: 100 μg/kg b.w. AFM1: 100 μg/kg b.w. Oral administration	14 days	Western blot analysis	AFB1: Decrease the protein level of TNF-α AFM1: Decrease the protein level of TNF-α	[100]
Pig (piglet, *n* = 24)	AFB1: 320 ppb Contaminated feed	30 days	ELISA	Decrease in the protein expression of IL-1β, IL-6, IFN-γ, TNF-α and in duodenum	[147]
Broiler (Cobb, *n* = 576)	AFB1: 40 μg/kg Contaminated feed	21 days	RT-PCR	Decrease in the transcript level of sIgA	[102]
Broiler chicks (Ross 308, *n* = 336)	AFs (AFB1+AFB2+AFG1+AFG2) 0.5 and 2 mg/kg feed Contaminated feed	28, 42 days	Histological analysis	Increase in the number and diameter of lamina propria lymphoid follicles in jejunum	[104]
Shrimp (*Litopenaeus vannamei*, *n* = 100)	AFB1: 5 ppm Contaminated feed	30 days	RNA-Seq	Identify 7 process or pathways related to immune system	[148]
Broiler chickens (Cobb 500, *n* = 240)	AFB1: 400 ppb Contaminated feed	21 days	Immunohistochemistry	Decrease in the CD4^+^ cells number in jejunum	[149]
Ochratoxin A					
Caco-2 cells	OTA: 5–45 μM	3, 12, 24 h	RT-PCR	Decrease in the mRNA expression of COX-2 and 5-LOX	[105]
Broiler chickens (*n* = 80)	OTA: 50 μg/kg b.w. Oral administration	21 days	RT-PCR	Increase in the transcript level of TNF-α and IL-1β in small intestine	[107]
Duck (White Pekin ducklings, *n* = 540)	OTA: 2 mg/kg Contaminated feed	21 days	ELISA	Increase in the protein level of TNF-α and IL-1β in jejunum	[150]
Deoxynivalenol					
IPEC-J2 cells	DON: 4 μM	24 h	RT-PCR	Increase in the mRNA expression of TNF-α and IL-8	[106]
IPEC-J2 cells	DON: 0.5–2.0 μg/mL	4, 8, 12 h	RT-PCR	Increase in the mRNA expression IL-1β, IL-6, COX-2, and TNF-α	[151]
IPEC-J2 cells	DON: 4 μM	12 h	Luminex multiplex assay	Increase in IL-6 and IL-8 protein level	[152]
Intestine 407 cells	DON: 25–1000 ng/mL	12 h	Luciferase assay, RT-PCR	Increase in IL-8 secretion and mRNA expression	[153]
IPEC-1 cells	DON: 1000 ng/mL	1 h	RT-PCR	Increase in IL-8 and MCP-1 mRNA expression	[154]
Pig (gilt, *n* = 72)	DON: 12 μg/kg b.w. Contaminated feed	7-42 days	Histological analysis	Increase in the lymphocytes number in intestine	[142]
Pig (growing pigs, *n* = 24)	DON: 3, 6, 12 mg/kg feed Contaminated feed	21 days	Histological analysis	Increase in the lymphocytes number in intestine	[141]
Fish (juvenile grass carp, *n* = 1440)	DON: 318–1515 μg/kg diet Contaminated feed	60 days	RT-PCR	Increase in the transcript level of pro-inflammatory cytokines (IL-1β, IL-6, IL-8, IL-12p35, IL-12p40, IL-15, IL-17D, TNF-α and IFN-γ) in intestine Decrease in the mRNA expression of anti-inflammatory cytokines (IL-10, IL-11, IL-4/13A, IL-4/13B and TGF-β1) in intestine	[138]
Pig (piglet, *n* = 24)	DON: 3.0 mg/kg Contaminated feed	35 days	RT-PCR	Increase the mRNA expression of IL-1β, IL-2, IL-6, IL-12p40, and MIP-1β in jejunum Increase the mRNA expression of IL-1β, IL-6, and TNF-α in ileum	[121]
Pig (piglet, *n* = 16)	DON: 3.5 mg/kg Contaminated feed	42 days	RT-PCR	Increase the mRNA expression of IL-4 and CXCL10 in jejunum Increase the transcript level of IFN-γ, IL-6, IL-8 and CXCL10 in ileum	[155]
Zearalenone					
IPEC-1 cells	ZEN: 25 μM	1 h	RT-PCR	No effect on the transcript level IL-1β, IL-6, IL-8, IL-17, and TNF-α Decrease in transcript level of IL-4 and IFN-γ	[156]
IPEC-1 cells	ZEN: 10–100 μM α-ZOL: 10–100 μM β-ZOL: 10–100 μM	24 h	ELISA	ZEN: a tendency to increase the secretion of IL-8 and IL-10 α-ZOL: decrease the secretion of IL-8 and IL-10 β-ZOL: decrease the secretion of IL-8 and IL-10	[118]
IPEC-1 cells	ZEN: 10 μM	24 h	RT-PCR	A tendency to increase the transcript level of IL-10, IL-18, CCL20 and MCP-1	[91]
IPEC-J2 cells	ZEN: 6, 8 μg/mL	24 h	RT-PCR	Increase in the transcript level of pro-IL-1β and pro-IL-18	[157]
Rat (Sprague-Dawley, *n* = 96)	ZEN: 1.3–146.0 mg/kg Contaminated feed	7 days	RT-PCR	Decrease in the mRNA expression of TNF-α and IL-1β in pregnant dams	[119]
Fumonisin B1					
HT-29 cells	FB1: 1.1–69.0 μM	48 h	Immunoenzymatic bioassay	No changes on IL-8 secretion	[93]
IPEC-1 cells	FB1: 2.6–100 μM	4 days	RT-PCR, ELISA	Decrease in the mRNA and protein expression of IL-8	[158]
Pig (piglet, *n* = 24)	FB1: 6 mg/kg Contaminated feed	35 days	RT-PCR	Increase the transcript level of IL-10 and IFN-γ in jejunum Increase the mRNA expression of TNF-α and IL-1β in ileum	[121]
Pig (*n* = 17)	FB1: 0.5 mg/kg b.w./d Gavage	7 days	RT-PCR	Decrease in the mRNA expression of IL-8 in the ileum	[158]
Citrinin					
Mice (BALB/c, *n* = 20)	CTN: 1, 5, and 10 mg/kg b.w. Gavage	14 days	Staining of immune cells for flow cytometric analysis	Increase CD8^+^ cells in intra-epithelial, Decrease CD19^+^ cells in intra-epithelial	[159]

**Table 7 toxins-12-00619-t007:** Modulation of the intestinal microbial barrier induced by mycotoxins.

Model	Dose/Administration Route	Exposure Time	Technique	Damage	References
Aflatoxin					
Broiler chicks (Ross 308, *n* = 336)	AFs (AFB1+AFB2+AFG1+AFG2) 0.5 and 2 mg/kg feed Contaminated feed	28, 42 days	Bacteriological examinations	Increase in the total negative bacteria, including *Escherichia coli*, *Salmonella*, *Klebsiella* in ileum	[104]
Broiler chicks (Ross 308, *n* = 336)	AFs (AFB1+AFB2+AFG1+AFG2) 0.5 and 2 ppm feed Contaminated feed	28, 42 days	Bacterial examinations	Increase in the total negative bacteria, including *Escherichia coli*, *Salmonella*, *Klebsiella* in ileum	[132]
Broiler (Cobb, *n* = 576)	AFB1: 40 μg/kg Contaminated feed	21 days	Bacterial examinations	No effect in the ileal bacteria populations, including *Lactobacilli*, *Bifidobacteria*, *C. perfringens*, *Escherichia coli*	[102]
Shrimp (Litopenaeus vannamei, *n* = 100)	AFB1: 5 ppm Contaminated feed	30 days	16S rRNA gene sequencing	Decrease in the types of intestinal microbiota Increase in the relative abundance of *Proteobacteria* and *Firmicutes* Decrease in the relative abundance of *Bacteroidetes*	[128]
Rat (Fischer 344, *n* = 20)	AFB1: 5, 25, 75 μg/kg b.w. Gavage	28 days	16S rRNA gene sequencing	Decrease in the fecal microbial diversity but increase evenness of community composition Decrease in the lactic acid bacteria number	[163]
Mice (Kunming, *n* = 24)	AFB1: 2.5, 4, 10 mg/L Gavage	60 days	16S rRNA gene sequencing	Decrease in the intestinal microbial diversity	[164]
Broiler Chickens (Cobb-Vantress, *n* = 480)	AFB1: 1, 1.5, 2 ppm Contaminated feed	21 days	Bacti flat bottom plate assay	Increase in the number of total gram-negative bacteria, total aerobic bacteria number, and total lactic acid bacteria in cecum	[165]
Ochratoxin A					
Caco-2 cells	OTA: 1–100 μM	12 h	Bacterial translocation assay	Increase in *Escherichia coli* translocation across monolayer	[166]
human intestinal microbial ecosystem	OTA: 2.5 μM	14 days	PCR-TTGE and PCR-DGGE	Alter gut microbiota diversity and cause the loss of beneficial species *Lactobacillus reuteri*	[167]
Rat (F344, *n* = 18)	OTA: 70, 210 μg/kg b.w. Gavage	28 days	16S rRNA sequencing, shotgun sequencing	Decrease in the diversity of the gut microbiota Increase in the relative abundance of *Lactobacillus*	[168]
Deoxynivalenol					
IPEC-J2 cells	DON: 4 μM	12 h	Bacterial translocation assay	Increase in *Escherichia coli* translocation across monolayer	[152]
IPEC-1 cells	DON: 5–50 μM	48 h	Bacterial translocation assay	Increase in *Escherichia coli* translocation across monolayer	[169]
Caco-2 cells	DON: 1-100 μM	12 h	Bacterial translocation assay	Increase in *Escherichia coli* translocation across monolayer	[166]
Mice (BALB/c, *n* = 42)	DON: 3.0 mg/kg Gavage	15 days	16S rRNA gene sequencing	Increase in the abundance of *Clostridiales*, *Lachnospiraceae* and *Blautia*	[140]
Mice (CD-1, *n* = 36)	DON: 1.0, 5.0 mg/kg Gavage	14 days	shotgun sequencing	Increase in the abundance of *Firmicutes* and *Bacteroidetes* in cecum	[170]
Broiler chickens (ROSS 308, *n* = 80)	DON: 2.5, 5 and 10 mg/kg diet Contaminated feed	35 days	16S rRNA gene sequencing	Increase in the abundance of *Firmicutes* in cecum Decrease in the abundance of *Proteobacteria* in cecum	[171]
Pig (piglet, *n* = 24)	DON: 2.5 mg/kg Contaminated feed	28 days	Capillary electrophoresis single-stranded conformation polymorphism	Increase in fecal aerobic mesophilic bacteria number	[172]
Rat (Wistar, *n* = 18)	DON: 60, 120 μg/kg b.w. Gavage	40 days	16S rRNA gene sequencing	Increase in the relative abundance of *Coprococcus* genus	[173]
Rat (Sprague-Dawley, *n* = 20)	DON: 100 μg/kg b.w. Gavage	28 days	RT-PCR	Increase in the concentration of *Bacteroides* and *Prevotella* genera Decrease in the expression *Escherichia coli*	[174]
Rat (Wistar, *N* = 80)	DON: 2, 10 mg/kg Contaminated feed	28 days	16S rRNA gene sequencing	No effect on the composition of the gut microbiota	[175]
Zearalenone					
Mice (BALB/c, *n* = 40)	ZEN: 10 mg/kg b.w. Gavage	14 days	16S rRNA gene sequencing	Decrease in the abundance of *Firmicutes* Decrease in the abundance of *Bacteroidetes*	[176]
Pig (gilt, *n* = 75)	ZEN: 40 μg/kg b.w. Oral administration	42 days	EcoPlate tests	Decrease in mesophilic aerobic bacteria number Decrease in the level of *Enterobacteriaceae* family	[177]
Fumonisin B1					
Pig (piglet, *n* = 24)	FBs (FB1+FB2): 11.8 ppm Contaminated feed	63 days	Capillary single-stranded conformation polymorphism analysis	Alter the digestive microbiota balance	[178]
Patulin					
Caco-2 cells	PAT: 1–100 μM	12 h	Bacterial translocation assay	Increase in *Escherichia coli* translocation across monolayer	[166]

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
