# Peer review of "The Compromised Intestinal Barrier Induced by Mycotoxins"

_toxins, 2020, doi:10.3390/toxins12100619_

Round 1

Reviewer 1 Report

This literature review presents the effects of the main mycotoxins on the barrier function of the intestine. The manuscript is clear, very well documented, and quite easy to read despite its length because of the many tables that summarize the results of the different studies reviewed.

This manuscript seems very interesting to me and I have only very minor comments to make :

- review the writing of the L11-14: not very well written compared to the rest of the manuscript.

- add T2 and HT2 toxins in Table 1, (these toxins are cited in the rest of the manuscript)

- L474: put "Lactobacillus plantarum" in italic characters.

- check the list of abbreviations

- avoid sentences that begin with "And".

Author Response

Responds to reviewers' comments:

Reviewer #1:

This literature review presents the effects of the main mycotoxins on the barrier function of the intestine. The manuscript is clear, very well documented, and quite easy to read despite its length because of the many tables that summarize the results of the different studies reviewed.

This manuscript seems very interesting to me and I have only very minor comments to make:

(1): review the writing of the L11-14: not very well written compared to the rest of the manuscript.

AU: Thanks for your suggestion. As your suggestion, we have modified the related description as “We also provide important information on deoxynivalenol, a leading mycotoxin implicated in intestinal dysfunction, and other adverse intestinal effects induced by other mycotoxins, including aflatoxins and ochratoxin A.” in the line of 11-14 in the revised manuscript.

(2): add T2 and HT2 toxins in Table 1, (these toxins are cited in the rest of the manuscript)

AU: Thanks for your suggestion. As your suggestion, we have added T-2 and HT-2 toxins in Table 1 in the line of 35 in the revised manuscript.

Table 1. The overview of mycotoxins.

Mycotoxin

Toxic effects

IARC classification

Health guidance value

References

AFB1,

AFM1

carcinogenic

Group 1

As low as reasonably achievable (ALARA principle)

[5,216]

OTA

nephrotoxic, teratogenic, immunotoxic, neurotoxic

Group 2B

TWI = 120 ng/kg bw/w

PTWI = 100 ng/kg bw/w

[5,216]

FB1

hepatotoxicity, nephrotoxicity

Group 2B

PMTDI (FB1+FB2 +FB 3) = 2 μg/kg bw/d

TDI (FB1+FB2 +FB3) = 2 μg/kg bw/d

[1,5,216]

DON

immunotoxic

Group 3

TDI = 1 μg/kg bw/d

[5,216-218]

ZEN

reproductive toxicity

Group 3

PMTDI = 0.5 μg/kg bw/d

TDI = 0.25 μg/kg bw/d

[5,216]

PAT

hepatotoxicity

Group 3

PMTDI = 0.4 μg/kg bw/d

[1,5,215]

NIV

immunotoxicity, hematotoxicity, myelotoxicity

Group 3

TDI = 1.2 μg/kg bw/d

[2,5,219]

T-2, HT-2

toxic on the skin and mucous membranes

Group 3

PMTDI (T-2+HT-2) = 0.06 μg/kg bw/d

TDI (T-2+HT-2) = 0.1 μg/kg bw/d

[5]

(3): L474: put "Lactobacillus plantarum" in italic characters.

AU: Thanks for your suggestion. As your suggestion, we have put “Lactobacillus plantarum in italic characters in the line of 467 in the revised manuscript as “Lactoferrin [75], grape seed [148], silymarin [137], yeast cell wall [109], Lactobacillus plantarum [107] and mannanoligosaccharides [118] have alleviated AFs induced intestinal disturbances in human intestinal epithelial cells, pigs, broiler chicks and mice.”

In addition, the title of this reference [107] also modified in the line of 824 of the revised manuscript.

(4): check the list of abbreviations

AU: Thanks for your suggestion. As your suggestion, we have added PMTDI, TDI, TWI, PTWI, CCK-8, RT-qPCR, TEER, ELISA, IFN-γ, MCP-1, COX-2 and CXCL10 in the list of abbreviations and organized them by alphabetical order in the line of 497-530 of the revised manuscript.

(5): avoid sentences that begin with "And".

AU: Thanks for your suggestion. As your suggestion, the whole manuscript has been checked by an English native speaker to avoid such description. We thank International Science Editing (http://www.internationalscienceediting.com) for editing this manuscript. According to these editing suggestions, we have modified carefully in the revised manuscript.

Reviewer 2 Report

Review on

„The compromised intestinal barrier induced by mycotoxins“

The manuscript reviews 220 relevant papers on the topic. The authors ordered the influence of mycotoxin on physical/chemical/immunological and microbial barrier. The utilized technique and model system is specified.

First an overview on the mycotoxins is given. Second the different barrier levels are depicted in a scheme (Fig.1). This diagram is very similar to the diagram of [16, Natividad et al. 2013]. In a third section the possible model systems are discussed. In the major part of the review the known effects and findings are given as tables and summarized in short texts. In section 5 and 6 the topics “leaky gut” and “interactive effects” are briefly discussed.

The structure of the review is clear and the tables are self-explanatory. As far as I could see, the data in the tables are correct and carefully extracted.

A useful tool in the field.

Minor remarks:

Abstract: …..And The current review….

1.Introduction

Please include TWI,PMTDI,TDI, PTWI in the abbreviation list.

  1. Components of intestinal barrier

….[21,22] (Furuse et al. 1993….

…immune cells in LP ….. abbreviation list or else

  1. Intestinal dysfunction induced by mycotxins

Tables: the position of the lines is confusing, please correct.

Table 3:  Fumonisin B1   ….FB1: 6_mg/kg

4.1.1. Effects of …… [97-99]. It has been reported that ZEN has have

4.2.1 ….. in IPEC-J2_cells [129].

4.3.2 ……revealed that 7 processes or pathways

4.4.1 ….  and PAT [158,159](

Page 30, line 65: In addition, The intestinal …

Author Response

Responds to reviewers' comments:

Reviewer #2:

The manuscript reviews 220 relevant papers on the topic. The authors ordered the influence of mycotoxin on physical/chemical/immunological and microbial barrier. The utilized technique and model system is specified.

First an overview on the mycotoxins is given. Second the different barrier levels are depicted in a scheme (Fig.1). This diagram is very similar to the diagram of [16, Natividad et al. 2013]. In a third section the possible model systems are discussed. In the major part of the review the known effects and findings are given as tables and summarized in short texts. In section 5 and 6 the topics “leaky gut” and “interactive effects” are briefly discussed.

The structure of the review is clear and the tables are self-explanatory. As far as I could see, the data in the tables are correct and carefully extracted.

A useful tool in the field.

Minor remarks:

(1): Abstract: …..And The current review….

AU: Thanks for your suggestion. As your suggestion, we have modified the description in line 11-14 in the revised manuscript as “We also provide important information on deoxynivalenol, a leading mycotoxin implicated in intestinal dysfunction, and other adverse intestinal effects induced by other mycotoxins, including aflatoxins and ochratoxin A”.

(2): Introduction

Please include TWI,PMTDI,TDI, PTWI in the abbreviation list.

AU: Thanks for your suggestion. As your suggestion, we have added PMTDI, TDI, TWI, PTWI, CCK-8, RT-qPCR, TEER, ELISA, IFN-γ, MCP-1, COX-2 and CXCL10 in the list of abbreviations and organized them by alphabetical order in the line of 497-530 of the revised manuscript.

(3): Components of intestinal barrier

….[21,22] (Furuse et al. 1993….

AU: Thanks for your suggestion. As your suggestion, we have deleted the description of references. The related contents have been modified as “TJs are the multi-protein complexes, and consist of different transmembrane proteins, e.g. junctional adhesion molecules (JAMs), claudins, occludin and zonula occludens (ZO) proteins [21,22]” in the line of 63-65 in the revised manuscript.

(4): …immune cells in LP ….. abbreviation list or else

AU: Thanks for your suggestion. The position LP first appeared in the manuscript is in line of 52 as “epithelial cells and immune cells in lamina propria (LP)”. Therefore, we directly used “LP” abbreviation in the line of 86 as “The immunological barrier comprises immune cells in LP (e.g. dendritic cells, resident macrophages, B cells and T cells)” in the manuscript. In addition, we checked that the abbreviation of “LP” existed in the abbreviation list in the manuscript.

(5): Tables: the position of the lines is confusing, please correct.

AU: Thanks for your suggestion. As your suggestion, we have modified the position of lines in Tables 3-7 in the revised manuscript.

(6): Table 3:  Fumonisin B1   ….FB1: 6_mg/kg

AU: Thanks for your suggestion. As your suggestion, we have modified the description as “FB1: 6 mg/kg” in the page 13 in the revised manuscript.

(7): 4.1.1. Effects of …… [97-99]. It has been reported that ZEN has have

AU: Thanks for your suggestion. As your suggestion, we have modified the description as “It has been reported that ZEN has no effect on TEER values of IPEC-1 cell” in the line 221-222 in the revised manuscript.

(8): 4.2.1 ….. in IPEC-J2_cells [129].

AU: Thanks for your suggestion. As your suggestion, we have modified the description as “no significant increases in β-defensin 1 and β-defensin 2 secretion occurred in IPEC-J2 cells [129]” in the line 283-284 in the revised manuscript.

(9): 4.3.2 ……revealed that 7 processes or pathways

AU: Thanks for your suggestion. As your suggestion, we have modified the description as “Transcriptome analysis revealed that 7 processes or pathways related to immune system were grouped in Litopenaeus vannamei as the result of AFB1 challenge [149]” in the line 337-338 in the revised manuscript.

 (10): 4.4.1 ….  and PAT [158,159](

AU: Thanks for your suggestion. As your suggestion, we have deleted the brackets and the related description was modified as “Caco-2 cell monolayers are induced by DON, NIV and PAT [158,159]” in the line of 368-369 in the revised manuscript.

(11): Page 30, line 65: In addition, The intestinal …

AU: Thanks for your suggestion. As your suggestion, we have modified the description as “The intestinal microflora may also affect mucus production” in the line of 424 in the revised manuscript.

The whole manuscript has been checked by an English native speaker to avoid such description. We thank International Science Editing (http://www.internationalscienceediting.com) for editing this manuscript. According to these editing suggestions, we have modified carefully in the revised manuscript.